# Characterization of the Glutamine Synthetase Gene Family in Wheat (*Triticum aestivum* L.) and Expression Analysis in Response to Various Abiotic Stresses

**DOI:** 10.3390/ijms26199403

**Published:** 2025-09-26

**Authors:** Zhiyong Zhang, Xiaojiao Zhang, Yanling Mu, Huali Wang, Lulu Wang, Furong Nai, Yihao Wei, Shuping Xiong, Xinming Ma, Huiqiang Li, Xiaochun Wang

**Affiliations:** 1College of Agronomy, Henan Agricultural University, Zhengzhou 450046, China; zhiyongzhang@henau.edu.cn (Z.Z.); zhang_max98@163.com (X.Z.); yanling_mu2023@163.com (Y.M.); hualiw2021@126.com (H.W.); wanglulu9501@163.com (L.W.); weiyihao@henau.edu.cn (Y.W.); shupxiong@henau.edu.cn (S.X.); maxinming@henau.edu.cn (X.M.); 2College of Life Sciences, Henan Agricultural University, Zhengzhou 450046, China; naifurong0407@163.com

**Keywords:** wheat, TaGS, gene family, expression analysis, abiotic stress

## Abstract

Glutamine synthetase plays an essential role in regulating plant growth and development. However, few studies have analyzed the roles of TaGS in wheat under abiotic stress conditions. In this study, we identified and analyzed the members of the TaGS gene family in *Triticum aestivum* L., focusing on their gene characteristics, phylogenetic evolution, cis-elements, transcriptional and post-translational modifications, and expression profiling in response to abiotic stress. Twelve TaGS genes were divided into four subfamilies. The synteny analysis revealed that wheat and the five other species share GS homologs. Several potential transcription factors were identified as regulators of TaGS genes. TaGS contains 19 microRNA binding sites, phosphorylation sites, and ubiquitination sites. TaGS genes exhibited tissue-specific expression across various developmental stages and were differentially expressed in response to abiotic stress. For instance, *TaGS1-3-4A/4B/4D* were upregulated in the leaves and roots of wheat seedlings under abiotic stress conditions. Furthermore, gene ontology annotation was performed on the TaGS-interacting proteins screened by immunoprecipitation–mass spectrometry to elucidate the regulatory network associated with TaGS. This study lays a foundation for further functional research of TaGS genes in response to abiotic stress and provides potential information for enhancing stress tolerance in wheat.

## 1. Introduction

Bread wheat (*Triticum aestivum* L.) is one of the most widely cultivated crops globally, providing approximately 20% of the calories and proteins in the human diet [1]. As the global population increases, the demand from consumers and businesses for wheat processing into various final products is also gradually increasing [2]. It is urgent to improve both the wheat yield and quality. The protein content and composition in the grains directly affect the viscoelastic properties of wheat dough as well as the quality of flour and food processing [3]. Nitrogen (N) application significantly increases wheat protein content, yield, and quality. However, excessive use of N fertilizer not only increases the cost but also leads to environmental pollution problems. Therefore, improving nitrogen use efficiency (NUE) is essential for sustainable agriculture. NUE is the combination of N uptake efficiency, N utilization efficiency, and N remobilization efficiency. A series of studies have shown that the glutamine synthetase (GS)/glutamate synthase (GOGAT) cycle plays a key role in NUE and exhibits a positive correlation with grain protein content [4]. GS catalyzes the conversion of ammonium (NH_4_^+^) and glutamate (Glu) into glutamine (Gln), while GOGAT catalyzes Gln and α-ketoglutarate to produce two molecules of Glu [5]. Both Glu and Gln play an important role in metabolism, growth, development, and signal transduction [6].

In higher plants, there are two types of GS isoenzymes: cytoplasmic GS1 and plastid GS2, which are located in the cytoplasm and plastid, respectively [7,8]. GS1 is predominantly localized in the vascular tissues of plants and is responsible for primary N assimilation and N remobilization. GS2 is mainly located in the photosynthetic tissues and contributes to the assimilation of NH_4_^+^ released from nitrate reduction in plastids and during photorespiration in plants [9]. The GS1 type subfamily has three to five members and is encoded by multiple nuclear genes [10]. Gln-synt_N and Gln-synt_C are the two main conserved domains in GS [11]. To date, the GS gene family has been identified and extensively studied in several monocotyledon and dicotyledon species, such as, wheat [12], Arabidopsis [13], common bean [14], alfalfa [15], and other species.

A previous study showed that the expression level of TaGS genes revealed significant differences in the roots of two wheat varieties with different NUEs, indicating that GS plays a key role in N metabolism [16]. Manipulating the expression of the GS genes can enhance crop growth, development, NUE, and yield. Overexpression of *TaGSr* (also named *TaGS1-2*) in Arabidopsis significantly improved N uptake and assimilation [17]. Overexpression of *TaGS1* in rice improved yield [18]. Overexpression of *HvGS1;1* in barley significantly improved NUE and yield compared with the wild type [19]. The *TaGS1-2* and *TaGS1-3* loci in wheat were associated with QTL for grain protein content, and overexpression of the *GS* gene improved roots, spike size, and N content [20,21]. *TaGS2-2Ab* transgenic lines exhibited an increased spike number, grain number per spike, thousand-grain weight, and high NUE [22].

During the growth and development stages of plants, they are inevitably exposed to various abiotic stresses, including heat, cold, drought, high salinity, metal toxicity, and their combinations. Plants have evolved physiological and biochemical processes to adapt or respond to external abiotic stress. Several studies have demonstrated that GS plays a role in response to various abiotic stresses in plants. In rice, GS enzymes are significantly induced, contributing to the regulation of intracellular reactive oxygen species (ROS) scavenging and energy metabolism under salt stress conditions [23]. The ectopic expression of the pine *GS1* gene in poplar exhibited higher expression levels of GS, GOGAT, and enhanced drought tolerance [24]. Díaz et al. suggested that LjGS2 was involved in proline biosynthesis and drought stress response in *Lotus japonicus* [25]. Overexpression of the *GS1* gene in tobacco enhanced the tolerance of transgenic plants to drought stress by negatively regulating water channel proteins [26]. Overexpressing the *GS2* gene in rice increased the tolerance of transgenic plants to salt and cold stresses [27]. However, there are few studies on the functions of TaGS genes in response to abiotic stress in wheat. In this study, we identified TaGS gene family members in the latest version of the wheat genomic data. A comprehensive analysis of the physicochemical properties, phylogeny, sequence characteristics, gene duplication events, and regulatory modules of the TaGS gene family members was conducted. Transcriptomic data and quantitative real-time PCR were used to determine the expression profiling of TaGS genes in wheat under various abiotic stress conditions. This study offers significant theoretical support for the functions of wheat TaGS genes and suggests potential applications for breeding wheat varieties with high stress tolerance.

## 2. Results

### 2.1. Identification and Characterization of TaGS Gene Family in Wheat

TaGS genes were identified in wheat using the six *Arabidopsis thaliana* GS protein sequences as query sequences for BLASTP search in the wheat protein database. Further screening was conducted on these candidate genes with the conserved GS domain using the SMART and Pfam databases. Finally, a total of 12 TaGS genes from wheat were identified. Phylogenetic classification and chromosome location were used to rename the TaGS genes (Table 1). The TaGS proteins, encoded by 12 TaGS genes, vary in length from 354 amino acids (TaGS1-2-4A/4B/4D) to 427 (TaGS2-2A/2B/2D) amino acids. The predicted molecular weight (MW) varies from 38.66 kDa (TaGS1-2-4D) to 46.70 kDa (TaGS2-2A/2D). The predicted isoelectric point (pI) of these TaGS proteins varies from 5.30 (TaGS1-3-4A) to 5.89 (TaGS2-2B), indicating that TaGS proteins exhibit weak acidity (Table 1). Members of the TaGS1 subfamily are localized in the cytoplasm, while TaGS2 members are localized in the chloroplasts, according to the prediction of protein subcellular localization (Table 1).

### 2.2. Evolutionary Relationship, Gene Structure, and Conserved Domains Analysis of GS

The divergence and evolutionary relationship of the GS gene family from wheat and other plant species were investigated. The phylogenetic analysis of the GS protein sequences from wheat and other representative species, including rice, maize, Arabidopsis, alfalfa, and common bean, was carried out using the MEGA version 12.0 software (https://www.megasoftware.net/dload_win_gui, accessed on 20 September 2025). The rootless phylogenetic tree was generated using the neighbor-joining method (Appendix A). The phylogenetic analysis showed that the TaGS gene family could be classified into two subfamilies: GS1 and GS2. GS1 divergence occurred before the differentiation of monocotyledons and dicotyledons. Because the GS1 protein sequences of monocotyledons (wheat, rice, and maize) and dicotyledons (Arabidopsis, common bean, and Alfalfa) differ significantly, they cannot be clustered in the same branch. The GS1 of the gramineous plants, such as wheat, maize, and rice, has three subgroups, among which the GS1-1 and GS1-3 subgroups are closely related. The GS2 subfamily of these six species is grouped in one branch and is conserved, indicating that GS2 has undergone convergent evolution in monocotyledons and dicotyledons (Figure 1a).

The evolution of gene families largely derives from the diversity of gene structures. Therefore, we analyzed the exon/intron organization of TaGS genes in the genome sequence of wheat. All members of the TaGS gene family consist of 9 to 14 exons/introns. Except for *TaGS2-2A*, most members also have 3′ untranslated regions (UTRs) and 5′ UTRs (Figure 1b). The numbers of exons of TaGS1-1, TaGS1-2, TaGS1-3, and TaGS2 subclades are 11, 9, 12, and 13, respectively. Genes within the same subclass of TaGS show similar numbers and orders of exons and introns.

The Simple Modular Architecture Research Tool (SMART) and Multiple Expectation Maximization for Motif Elicitation (MEME) websites were analyzed for the domains and conserved motifs of TaGS proteins (Figure 1c,d). The N-terminal (Gln-synth_N) and the C-terminal conserved domains (Gln-synth_C) were found in all GS members except ZmGS1-5, which only had the Gln-synth_C domain, potentially affecting its catalytic function. In addition to the conserved Gln-synth_N and Gln-synth_C domains in TaGS proteins, a total of 15 conserved motifs were detected in TaGS based on MEME analysis (Appendix A). Members of the same subfamily in the phylogenetic tree shared similar motifs, which are remarkably distinct from those of other subgroups. Motif 1–motif 9 were highly conserved among all TaGS proteins, while motif 14 was specific to the GS1 subfamily, motif 12 and motif 13 were uniquely detected in the GS2 subfamily. Motif 15 was unique to the GS2 subfamily in wheat, rice, and maize, likely distinguishing monocotyledons from dicotyledons. For plastids GS2-2A/2B/2D, the N-terminal 60 amino acids were essential for its localization, while the C-terminal 16 amino acids were essential for enzymatic activity [28] (Appendix A).

### 2.3. Synteny and Ka/Ks Analysis of TaGS

Gene duplication events are one of the key mechanisms for organisms to acquire new genes and generate genetic novelty, facilitating plant evolution by enhancing adaptability to extreme environments [29]. Whole-genome duplication in the A, B, and D subgenomes of hexaploid wheat resulted in the dispersion of homologs, according to the analysis of TaGS homologous gene pairs (Appendix A). As a result of fragmental replication, *TaGS1-1-6A/6B/6D* were located in the distal region of chromosome 6, *TaGS2-2A/2B/2D* were located in the distal region of chromosome 2, and *TaGS1-2-4A/4B/4D* and *TaGS1-3-4A/4B/4D* were located in the distal region of chromosome 4. These results indicated that the expansion and evolution of the wheat TaGS genes were due to whole-genome replication and fragmental replication.

The synteny relationships of GS genes in the wheat and five other species were analyzed; the results showed 12, 12, 3, 1, and 0 homologous gene pairs between wheat TaGSs and GS genes in rice, maize, common bean, Arabidopsis, and alfalfa, respectively (Figure 2a; Appendix A). Wheat GS exhibited the highest similarity to the GS of rice and maize. Homologous genes were distributed on chromosomes 2A, 2B, 2D, 4A, 4B, 4D, 6A, 6B, and 6D. The GS homologous genes were distributed on chromosomes 4A, 4B, and 4D in common bean. The presence of *TaGS1-3-4A* (*TraesCS4A02G26690.1*) in wheat, rice, maize, bean, and Arabidopsis co-linear gene pairs suggests that this gene is widespread in both monocotyledonous and dicotyledonous plants. It likely emerged early in the evolutionary timeline, before species divergence, and plays a significant role in the GS family. The affinity of the GS gene among the six species was ordered as follows: maize, wheat, rice, bean, Alfalfa, and Arabidopsis, indicating that the evolution of GS genes varies between monocotyledonous and dicotyledonous plants (Figure 2b).

The effects of selective pressure and evolutionary factors on gene duplication and species evolution events were analyzed using natural selection analysis [30]. The selective pressure of duplication events was assessed using the ratio of nonsynonymous substitutions (Ka) and synonymous substitutions (Ks). A Ka/Ks ratio >1 represents positive selection, Ka/Ks = 1 indicates neutral selection, and Ka/Ks <1 indicates purifying selection. In the present study, we examined the Ka/Ks ratios between GS homologous gene pairs in wheat and other representative species (Figure 2c; Appendix A), and the results showed that the Ka/Ks ratios of TaGS homologous gene pairs ranged from 0 to 0.17. With a nonsynonymous mutation rate of 0, *TaGS2-2A/TaGS2-2D* and *TaGS1-1-6B/TaGS1-1-6D* underwent neutral genetic drift. The Ka/Ks ratios for all GS gene pairs were <1, with null values for common bean, wheat, and rice, due to significant sequence differences between monocotyledons and dicotyledons and their long genetic distances. These findings indicated that the GS gene pairs underwent purifying selection during evolution.

### 2.4. Analysis of cis-Acting Elements in Promoters of TaGS Gene

The promoter is essential for driving gene transcription. To further investigate the functions of the TaGS gene family, the 2 kb sequences upstream of the ATG start codon of the TaGS genes were extracted from the genome and analyzed for cis-acting elements in the promoter of TaGS genes using the PlantCARE online database (https://bioinformatics.psb.ugent.be/webtools/plantcare/html, accessed on 20 September 2025). The results showed that six categories of cis-acting elements involved in plant development, phytohormone responses, stress, light, promoter-associated functions, and transcription factor (TF)-binding sites were found in the promoter of TaGS genes (Figure 3 and Appendix A). Tissue-specific expression, protein metabolism, and cell cycle regulation are regulated by development-related response elements like the CAT-box, CCGTCC-box, O2-site, and MSA-like. Hormone-responsive elements are abundant in TaGS genes, including abscisic acid-responsive elements (ABRE, ABRE3a, and ABRE4), auxin-responsive elements (AuxRR-core and TGA-element), gibberellin-responsive elements (GARE-Motif, P-box, and TATC-box), jasmonic acid-responsive elements (JERE, CGTCA-Motif, AAGAA-Motif, and TGACG-Motif). All TaGS members have stress response elements (STREs) associated with heat stress, osmotic stress, low pH stress, cold stress (low-temperature responsive elements, LTR), defense and stress response (TC-rich repeats), anaerobic stress (ARE and GC-Motif), drought response (DRE core, MBS, and Myb), and injury stress. Light-responsive elements, including the G-box, Sp1, GT1-motif, and TCCC-motif, are present in all TaGS genes. The 12 TaGS gene promoters contain multiple binding sites for the MYB and MYC transcription factors. The presence of these cis-acting elements in the TaGS gene promoters suggested that the TaGS gene family may play an important role in plant growth, development, and stress responses in wheat.

Transcription factors modulate gene expression by binding to the promoters of their target genes. The regulatory network for the TaGS gene promoter was predicted using the PlantRegMap online website. A total of 1236 potential transcription factors may regulate 12 TaGS genes, including several TF families such as ERF (948), MYB (76), LBD (68), C2H2 (34), BBR-BPC (21), NAC (15), bZIP (6), AP2 (3), WRKY (2), bHLH (2), and Dof (1) (Appendix A).

### 2.5. MiRNA Target Site Analysis of TaGS Gene Family

MicroRNAs (miRNAs) are small non-coding RNAs that suppress gene expression of target genes by cleaving messenger RNAs (mRNAs) or inhibiting their translation. miRNAs play a role in multiple essential biological processes, such as growth, development, and tolerance to abiotic stress. A previous study suggested that *TaGS1-3-4B/4D* was regulated by tae-miR9677, which was specifically expressed in developing wheat grains [31]. In this study, we analyzed the post-transcriptional modification of miRNA in TaGS genes. Our results indicate that 8, 4, 5, and 5 miRNAs target *TaGS1-1*, *TaGS1-2*, *TaGS1-3*, and *TaGS2* genes, respectively (Figure 4; Appendix A). miRNAs interact with TaGS by blocking and cleaving transcripts. tae-miR9666 and tae-miR9773 inhibit translation of TaGS, while other miRNAs regulate TaGS translation by cleaving transcripts. The regulatory network between miRNAs and TaGS genes is shown in Figure 4. For instance, tae-miR9657-3p and tae-miR9666 target the *TaGS1-1-6A/6B/6D* genes. tae-miR164, tae-miR5384a, and tae-miR9664-3p target the *TaGS1-2-4A/4B/4D* genes, and tae-miR319 targets *TaGS2-2A/2B/2D* genes. *TaGS1-3-4A* was regulated by tae-miR1136 and tae-miR9773. These findings suggested that the expression of TaGS genes can be regulated by various miRNAs.

### 2.6. Analysis of Post-Translational Modification Sites of TaGS

Post-translational modifications (PTMs) enable a protein to perform various physiological functions and participate in multiple biological processes, such as protein synthesis and degradation, signal transduction, and stress response [32,33]. Six post-translational modifications of TaGS proteins were predicted, including phosphorylation, glycosylation, S-nitrosylation, palmitoylation, ubiquitination, and SUMOylation (Appendix A). With the exception of TaGS1-1, all TaGS proteins contain ubiquitination, phosphorylation, S-nitrosylation, and palmitoylation sites. Apart from TaGS1-3, all TaGS proteins have a glycosylation site. TaGS1-1 and TaGS1-3 have a SUMO site. These findings indicate that TaGS proteins might be regulated by various PTMs, potentially affecting their structure, function, localization, and activity.

### 2.7. Expression Profiles of TaGS

The expression patterns of TaGS genes were analyzed in five tissues (root, stem, leaf, spike, and grain tissues) across developmental stages and various abiotic stress conditions using Expression Visualization and Integration Platform (ExpVIP) RNA-seq data (Figure 5; Appendix A). The results revealed that TaGS genes were expressed differently across tissues at various developmental stages. For instance, *TaGS1-1-6A/6B/6D* and *TaGS1-2-4A/4B/4D* genes were highly expressed in roots, stems, leaves, spikes, and grains during the seedling, vegetative, and reproductive stages (Figure 5a). The highest expression levels of *TaGS1-1-6A/6B/6D* and *TaGS1-2-4A/4B/4D* were observed in roots during the vegetative and reproductive stages, and in grains at 2 days after anthesis (daa), but these genes were significantly downregulated in grains at both the 14 daa and 30 daa stages (Figure 5a). *TaGS1-3-4A/4B/4D* were predominantly expressed in spikes at 14 daa and in grains at the 30 daa stage. *TaGS2-2A* and *TaGS2-2D* exhibited higher expression in stems at anthesis and in leaves at the seedling and vegetative stages, while *TaGS2-2B* was lowly expressed in these tissues during developmental stages (Figure 5a).

To investigate the role of TaGS in response to abiotic stress, we analyzed the expression levels of the TaGS gene family under various abiotic stress conditions, including cold, drought, heat, and a combination of drought and heat, using publicly available RNAseq data from the expVIP database (Figure 5b). The results indicated that expression levels of *TaGS1-2-4A/4B/4D* and *TaGS1-3-4A/4B* were increased under cold (4 °C) stress conditions, while the expression levels of *TaGS1-1-6D* and *TaGS2-2A/2D* decreased. *TaGS2-2A* and *TaGS2-2D* were downregulated after 6 h of drought treatment and upregulated after 6 h of drought combined with heat treatment. *TaGS1-3-4B* was upregulated after 6 h of drought treatment, heat treatment, and drought combined with heat (Figure 5b). Under heat conditions, the expression levels of *TaGS1-1-6A/6B/6D* and *TaGS1-3-4B* at 6 h were higher than those at 1 h. In contrast, *TaGS1-2-4A/4B/4D* transcripts showed moderate downregulation at 6 h compared with 1 h of heat treatment. However, no significant changes were observed in *TaGS1-3-4A/4D* and *TaGS2-2B* transcripts under these abiotic stresses. These findings indicated that the TaGS gene family may play crucial roles in abiotic stress tolerance in wheat.

### 2.8. Expression Profiling of TaGS Genes Under Various Abiotic Stresses Using Quantitative RT-PCR Analysis

A variety of abiotic stresses, such as heat, drought, cold, flooding, and salinity, adversely affect crop growth, development, and yields, posing a threat to global food security. To further validate the response of the TaGS gene family to abiotic stress, the dynamic expression levels of 12 TaGS gene members in wheat roots and leaves were examined under NaCl, drought, heat, and cold stress conditions using qRT-PCR (Figure 6 and Figure 7).

Under salt conditions, *TaGS1-1-6B/6D* and *TaGS2-2A/2B/2D* showed significant downregulation in both leaves and roots. *TaGS1-2-4A/4B/4D* expression significantly decreased in leaves at 3 h and 6 h, as well as in roots. The expression levels of *TaGS1-3-4A/4B/4D* increased in both leaves and roots. *TaGS1-1-6A* was downregulated in roots and exhibited a minimal change in leaves.

Under drought conditions, *TaGS1-1-6B* and *TaGS2-2A/2B/2D* were significantly downregulated in both leaves and roots. *TaGS1-2-4A/4D* expression was only increased in leaves at 48 h. The expression of *TaGS1-2-4B* significantly increased in leaves at 12 h and 48 h. The expression of *TaGS1-3-4A/4B/4D* increased in leaves and roots. *TaGS1-1-6A* was upregulated in roots at 12 h. *TaGS1-1-6B* and *TaGS1-2-4A/4B/4D* decreased in roots. *TaGS1-1-6D* was downregulated in roots at 24 h and in leaves.

Under heat conditions, *TaGS1-1-6A/6B*, and *TaGS2-2B/2D* were significantly downregulated in leaves and roots. *TaGS1-1-6D* was upregulated in leaves at 6 h and in roots. *TaGS1-2-4A/4B/4D* expression decreased in leaves at 3 h and 6 h and then increased at 12 h and 24 h in leaves. *TaGS1-2-4A/4B/4D* expression decreased in roots. The expression of *TaGS1-3-4A/4B/4D* significantly increased in leaves, whereas *TaGS1-3-4D* was upregulated in leaves and downregulated in roots at 3 h. The expression of *TaGS1-3-4A* and *TaGS1-3-4B* was upregulated in roots at 24 h and 12 h, respectively. *TaGS2-2A* was downregulated in roots and exhibited a minimal change in leaves.

Under cold conditions, the expression of *TaGS1-3-4A/4B/4D* increased in leaves and roots. The expression of *TaGS2-2A/2B/2D* decreased in leaves and roots. *TaGS1-2-4A/4B/4D* were upregulated in leaves but downregulated in roots. *TaGS2-2A/2B/2D* was downregulated in leaves and roots. *TaGS1-1-6A/6B* was downregulated in leaves and roots. *TaGS1-1-6D* was downregulated in leaves at 6 h, 12 h, and 48 h but upregulated in roots at 48 h. In summary, the varied expression profiles of the TaGS gene under abiotic stress conditions indicated that they were involved in the abiotic stress response.

Furthermore, we analyzed the transcription activity of the *TaGS1-2-4D* promoter in Arabidopsis transgenic plants harboring *proTaGS1-2-4D::GUS* leaves under salt, heat, drought, and cold stress conditions (Appendix A). Compared with the control, the expression level of *TaGS1-2-4D* was downregulated under salt and heat stress (Appendix A), and the expression level of *TaGS1-2-4D* was upregulated under drought and cold stress conditions (Appendix A). These results were consistent with the qRT-PCR data, indicating the presence of cis-acting elements associated with various abiotic stresses within the promoter of *TaGS1-2-4D.*

### 2.9. Screening of TaGS-Interacting Proteins Using Immunoprecipitation–Mass Spectrometry

Previous studies suggested that *TaGS1-2*, *TaGS1-3,* and *TaGS2-2Ab* were associated with wheat grain protein content and thousand-grain weight [20,21,22]. To verify whether TaGS forms complexes with other proteins to perform biological functions in wheat grain during grain-filling stages, 210 potential TaGS-interacting proteins in the grains were identified through an immunoprecipitation–mass spectrometry (IP-MS) assay (Appendix A). These proteins were categorized into three groups based on their GO functional annotations: molecular function (MF), cellular component (CC), and biological process (BP) (Appendix A). Gene Ontology (GO) annotation was performed to investigate the potential functions of TaGS in biological processes.

TaGS-interacting proteins were annotated with 1170, 320, and 266 annotations for GO-BP, GO-MF, and GO-CC, respectively. The majority of the 17 highly enriched annotations in GO-MF were in the categories of catalytic activity (GO:0003824), ion binding (GO:0043167), nucleotide binding (GO:0000166), protein binding (GO:0017111), and hydrolase activity (GO:0016818). The 15 highly enriched annotations in GO-CC primarily involve organelle (GO:0043228), cytoplasm (GO:0005737), membrane (GO:0016020), and nucleus (GO:0005634) components. In GO-BP, there are 27 highly enriched entries, most related to organic matter metabolism (GO:0071704), cellular metabolism (GO:0044237), nitrogen metabolism (GO:0006807), response to stimulation (GO:0050896), response to stress (GO:0006950), and biosynthesis (GO:0009058). In summary, TaGS likely interacts with various proteins to form complexes involved in metabolism, biosynthesis, and responses to environmental stress.

## 3. Discussion

Glutamine synthetase, a key enzyme in N metabolism in plants, plays an essential role in regulating plant growth, development, and yield. Some studies have demonstrated that GS plays a role in response to abiotic stresses in plants [23,24,25,26,27]. However, limited studies have examined GS in wheat’s response to various abiotic stresses. In this study, 12 TaGS genes were identified and evenly distributed across the wheat A, B, and D subgenomes; each subgenome consisted of four GS genes, which were comparable to those of five in common bean [34] and Alfalfa [15], and six in Arabidopsis [13] and maize [35]. Phylogenetic analysis indicated that the TaGS gene family can be categorized into four subfamilies: TaGS1-1, TaGS1-2, TaGS1-3, and TaGS2. The physicochemical properties of TaGS proteins, such as amino acid number, molecular weight, and isoelectric point, differ among subgroups, indicating that genetic diversity of TaGS genes occurred during evolution. Gene duplication is the primary evolutionary mechanism that increases genetic complexity and diversity to improve the adaptation of plants to various environments [36]. The expansion of TaGS genes in wheat resulted from whole-genome duplication and segmental duplication events. Duplications of these GS genes contribute to the generation of novel biological functions and might improve the adaptation to environmental stimuli. The collinearity analysis of multiple species revealed that dicotyledons have more GS synteny gene pairs than monocotyledons, which is consistent with their shared evolutionary history. The Ka/Ks ratio of GS gene pairs was less than one across all species, demonstrating that TaGS genes have undergone purifying selection pressure (Figure 2c).

The number and structural characteristics of exons and introns directly affect the conservation and differentiation of homologous gene functions. All five species, including the GS members of wheat, have 9 to 14 exons and introns (Figure 1). The exon number and position of GS are relatively conserved in each subfamily. A previous study suggested that introns could delay the TaGS2 regulatory response [37]. The number of introns of TaGS2 is more than that of TaGS1; the result may affect the regulatory response of TaGS. There are two conserved domains and fifteen conserved motifs in the GS protein. The gene structure is diverse across six species, providing a biological basis for its different functions.

Various signals and environmental factors, such as cytokinin, auxin, cold, salt, drought, light, NH_4_^+^, and hydrogen peroxide (H_2_O_2_) [33,38], are linked to the numerous response elements in the promoter, regulating gene expression. Most promoters are constitutive, tissue-specific, or inducible; they can enhance plant stress tolerance by regulating the expression of stress response genes [39]. Transcription factors can bind to specific cis-acting elements in the promoter to activate or deactivate GS gene expression. Previous studies have shown that rice OsMYB55 binds to the *OsGS1;2* promoter, regulating the expression of GS enzymes, enhancing amino acid metabolism, and increasing resistance to heat stress [40]. The R2R3-MYB transcription factor directly binds to the AC-II element of the *Pinus sylvestris GS1b* promoter, activating *GS* gene transcription [41]. TaNAC2-5A enhanced N uptake and assimilation by directly regulating the expression of *TaGS2-2A* in wheat roots [42]. In this study, the 2.0 kb promoter regions upstream of each TaGS gene were analyzed to identify predicted cis-acting elements. Several cis-acting elements were predicted in the promoters of 12 TaGS genes, including ERF, MYB, LBD, C2H2, BBR-BPC, NAC, and bZIP (Figure 3 and Appendix A). These cis-acting elements regulate plant growth and development, hormone response, and stress responses [43]. These findings indicated that TaGS genes might be regulated by various transcription factors associated with plant stress response.

MicroRNAs play a significant role in development and stress response in plants by regulating the expression of target genes. Previous studies suggested that miR319 and miR164 could regulate leaf morphology and flower organ development by regulating TCP and NAC transcription factors, respectively. miRNAs are also involved in the regulatory processes of various abiotic stresses in rice, wheat, maize, and Arabidopsis [44,45,46]. In this study, our findings showed that 12 *TaGS* genes were targeted by a total of 19 miRNAs (Figure 4). For instance, *TaGS2-2A/2B/2D* were targeted by tae-miR164 and tae-miR319. Another miRNA also regulates TaGS genes. Therefore, our investigation offers evidence that some miRNAs regulate the expression levels of TaGS genes in wheat.

Post-translational modifications, such as phosphorylation, sumoylation, and ubiquitination, play key roles in various plant stress responses and signaling transduction [47,48,49]. GS activity is also regulated at the post-translational level in many species, such as Arabidopsis, maize, and tobacco [50,51,52]. For example, the phosphoserine-binding proteins 14-3-3 inactivate GS2 by phosphorylating Ser 97 [53]. Calcium-dependent protein kinase 28 (CPK28) phosphorylates GS2 at the Ser 334 and Ser 360 sites, mediating plant growth and defense responses in tomato, respectively [54]. Ubiquitin-like proteins bind to GS2 and GS1 polypeptides to regulate the activity of GS in Alfalfa [55]. Multiple phosphorylation and ubiquitination sites were identified in 12 TaGS proteins (Appendix A); these modifications may enable TaGS proteins to affect plant growth, development, and stress responses.

A few studies have demonstrated that GS plays a role in response to abiotic stresses in wheat. For example, GS1-3 was expressed in roots, mesophyll cells, and chloroplasts, enabling rapid assimilation of NH_4_^+^ into Gln under high-NH_4_^+^ stress conditions and avoiding ammonium toxicity in plants [56,57,58]. Overexpression of TaGS1 in tobacco transgenic plants exhibited enhanced tolerance to drought [26]. In this study, transcriptomic data and qRT-PCR analysis showed that the expression levels of the TaGS gene family members in leaves and roots of wheat were regulated under various abiotic stress conditions, such as cold, heat, salt, and drought (Figure 5, Figure 6 and Figure 7).

Previous studies suggested that GS forms complexes with other proteins to regulate various biological processes. For instance, ACR11, a uridylyltransferase-like protein, interacted with AtGS2 in vitro and in vivo. ACR11 acts as an activator of GS2, enhancing GS2 activity and improving N assimilation in Arabidopsis [59]. The conserved C-terminal domain of nodulin 26 (NOD26) could directly bind to GS1β1, forming a protein–protein complex that promotes the rapid assimilation of NH_4_^+^ and prevents ammonia toxicity [60]. TaGS1;1, TaGS1;3, and TaGS2 were expressed in wheat grains [61]. Therefore, we screened potential TaGS-interacting proteins in grain during the grain-filling stages using IP-MS. GO enrichment analysis revealed that these proteins exhibited binding, catalytic, and hydrolase activities and were involved in various biological processes, such as N metabolism and environmental response.

## 4. Materials and Methods

### 4.1. Identification and Analysis of TaGS Gene Family in Wheat

Nucleotide sequences, coding sequences (CDSs), protein sequences, and genome annotation general feature format version 3 (GFF3) files for wheat, maize, rice, common bean, Arabidopsis, and Alfalfa were downloaded from the EnsemblPlants database (http://plants.ensembl.org/index.html, accessed on 20 September 2025) (Appendix A). The GS protein sequences of wheat, maize, rice, common bean (*Phaseolus vulgaris* L.), and Alfalfa (*Medicago sativa* L.) were compared using BLASTP (version BLAST+ 2.17.0) with the threshold set to 1 × 10^−25^, using the six Arabidopsis AtGS protein sequences downloaded from the TAIR (https://www.arabidopsis.org/, accessed on 20 September 2025) database as query sequences. Duplicates were removed from the results, and the longest protein sequence for each ID was retained for subsequent analysis. Pfam (http://pfam.xfam.org/, accessed on 20 September 2025) and SMART (http://smart.emblheidelberg.de/, accessed on 20 September 2025) were used to search for GS proteins with the Gln-synt_C and Gln-synt_N conserved domains. A total of 38 GS protein sequences were identified, including 12 members of the wheat TaGS gene family (Appendix A).

### 4.2. Physicochemical Characterization and Subcellular Localization Analysis of TaGS

Expasy ProtParam web servers (http://web.expasy.org/protparam/, accessed on 20 September 2025) were used to determine the physicochemical properties of GS proteins, including molecular weight, isoelectric point, and amino acid number, and information on the chromosomal location of TaGS genes from the GFF3 genome annotation file. WoLF PSORT (https://wolfpsort.hgc.jp/, accessed on 20 September 2025) predicted the subcellular localization of GS proteins.

### 4.3. Phylogenetic Relationships, Gene Structure, and Conserved Domain Analysis

Multiple protein sequence alignment of GS was performed using ClustalX (version 2.1), and a phylogenetic tree analysis based on neighbor-joining was subsequently generated using the MEGA X software [62]. Bootstrap values were generated with 1000 replicates. The evolutionary tree was improved by Evolview (https://www.evolgenius.info/evolview-v2/, accessed on 20 September 2025). The Multiple Expectation Maximization for Motif Elicitation (MEME) online program (http://meme-suite.org/index.html, accessed on 20 September 2025) was used to identify conserved motifs in the GS protein, with the maximum parameter set to 15. The GS gene structure, including introns, exons, and untranslated regions, was visualized using the Tbtools software (version 2.311).

### 4.4. Analysis of Synteny, Non-Synonymous and Synonymous (Ka/Ks)

Gene duplication events within the wheat TaGS and other GS gene homology were analyzed based on nucleotide sequence files and genome annotation files using the MCScanX tool (https://github.com/wyp1125/MCScanX, accessed on 20 September 2025) in the Tbtools software. The Simple-Ka/Ks Calculator tool was used to compare the codon sequences of homologous pairs and determine the ratio of non-synonymous (Ka) and synonymous (Ks) substitutions based on the CDS, protein sequences, and homologous pairs acquired from the synteny analysis.

### 4.5. Analysis of Promoters, Transcription Factors, and MicroRNA (miRNA) Targets

The cis-acting element analysis of promoter regions (the 2.0 kb sequence upstream of the start codon) of the 12 TaGS genes was carried out using the PlantCARE software (version 7.6.9) program (http://bioinformatics.psb.ugent.be/webtools/plantcare/html/, accessed on 20 September 2025), and it was visualized using the Simple BioSequence Viewer (version 2.1.19) in Tbtools. PlantRegMap (http://plantregmap.gao-lab.org/binding_site_prediction.php, accessed on 20 September 2025) predicted the transcription factors for the target gene using the promoter sequences of the TaGS gene family.

psRNATarget (http://plantgrn.noble.org/psRNATarget/, accessed on 20 September 2025) was used to predict the interacting miRNAs of the DNA sequences of the TaGS genes. The Cytoscape software (version 3.30.4) was then used to visualize the results [63].

### 4.6. Prediction of Post-Translational Modification Sites of TaGS

The ubiquitination sites, phosphorylation sites, and glycosylation sites of TaGS proteins were predicted using the Ubicomb (http://nsclbio.jbnu.ac.kr/tools/UbiComb/, accessed on 20 September 2025), NetPhos2.0 Server (http://www.cbsdtu.dk/services/NetPhos/, accessed on 20 September 2025), and DTU Health Tech (https://www.healthtech.dtu.dk/english, accessed on 20 September 2025), respectively. The S-nitrosylation, palmitoylation, and SUMOylation sites of TaGS proteins were analyzed using MusiteDeep (https://www.musite.net/, accessed on 20 September 2025) [64].

### 4.7. Expression Profiling of TaGS Gene Family

expVIP (http://www.wheat-expression.com/, accessed on 20 September 2025) was utilized to collect RNA-seq data of TaGS genes across various developmental stages, tissues, and abiotic stress conditions. Expression profiling was then performed on the gene expression data.

### 4.8. Plant Materials, Growth Conditions, and Stress Treatments

The seeds of the common wheat cultivar Yumai49 (YM49) in this study were produced by the Pingan Seeds company. To analyze the expression levels of TaGS genes under various abiotic stress conditions, the YM49 seeds were surface-sterilized using a 1% hydrogen peroxide solution for 10 min and rinsed five times with sterile distilled water. They were then placed on moistened filter paper in Petri dishes and germinated for two days at 28 °C in the dark. Fifteen wheat seedlings were transferred into a hydroponic device containing 1 L of Hoagland’s nutrient solution (pH 6.0) in a growth incubator at 22 °C and 65% humidity with long-day conditions (16 h light/8 h dark). Hoagland’s nutrient solution was replaced every 3 days. After two weeks, the seedlings were transferred to heat stress (40 °C), cold stress (4 °C), salt stress (Hoagland’s nutrient solution containing 200 mM sodium chloride), and drought stress [Hoagland’s nutrient solution containing 20% m/V polyethylene glycol 6000 (PEG 6000)] conditions for treatment. The leaves and roots of the seedlings were harvested at 0 h, 3 h, 6 h, 12 h, 24 h, and 48 h after treatment, respectively. The leaves and roots of the seedlings were then immediately frozen with liquid nitrogen and then stored at −80 °C until used for subsequent RNA extraction.

### 4.9. RNA Extraction and Complementary DNA (cDNA) Synthesis

The total RNA from the leaves and roots of wheat seedlings under four abiotic stress conditions was extracted using TRIzol reagent. The quantity of the isolated RNA was determined using a NanoDrop 2000 spectrophotometer (Thermo Fisher Scientific, Waltham, MA, USA), and the integrity of the total RNA was confirmed by electrophoresis in 1% agarose gel. The RNA was reverse-transcribed into cDNA using the HiScript III; 1st strand cDNA synthesis Kit (Novozymes, Nanjing, China) according to the manufacturer’s protocol.

### 4.10. Quantitative Real-Time PCR Assay and Statistical Analysis

The gene expression analysis was performed by quantitative real-time PCR (qRT-PCR) using an ABI StepOnePlus Real-Time PCR system (Thermo Fisher, USA). The 20 μL PCR reaction volume contained 10 μL of 2× RealStar Green Fast Mixtures, 0.5 μL each of the forward (F)/reverse (R) primer (10 μmol L^−1^), 0.5 μL of the diluted cDNA sample (100 ng), and 8.5 μL of RNase-free water. The PCR reaction conditions were pre-denaturation at 95 °C for 5 min, followed by 40 cycles at 95 °C for 15 s and 62 °C for 15 s. Three biological replicates and three technical replicates were carried out. Using *TaACTIN* as the internal reference control, the relative expression level of genes was calculated with the 2^−ΔΔCT^ method. GraphPad Prism 10.1.2 was used to create the graphs. The specific primers for qRT-PCR were designed using the Primer Premier 5.0 software and are listed in Appendix A. The qRT-PCR data were analyzed using Microsoft Excel, version 16. Statistical differences were determined using one-way ANOVA. * *p* < 0.05, ** *p* < 0.01, *** *p* < 0.001, and **** *p* < 0.0001.

### 4.11. Histochemical β-Glucuronidase Staining Assays and Microscopy

Histochemical assays of β-glucuronidase (GUS) activity were conducted as described previously [17]. The leaves of forty-day-old Arabidopsis transgenic plants harboring *proTaGS1-2-4D::GUS* seedlings were collected and transferred to control (½ MS liquid medium, 22 °C), heat stress (½ MS liquid medium, 40 °C), cold stress (½ MS liquid medium, 4 °C), salt stress (½ MS liquid medium containing 200 mM sodium chloride, 22 °C), and drought stress (½ MS liquid medium containing 20% m/V PEG 6000, 22 °C) for 6h. These samples were incubated in GUS solution [50 mM sodium phosphate buffer (pH 7.2), 10 mM EDTA (pH 8.0), 0.5 mM K_4_Fe(CN)_6_∙3H_2_O, 0.5 mM K_3_Fe(CN)_6_, 0.1% Triton X-100, and 2 mM 5-bromo-4-chloro-3-indolyl-beta-D-glucuronide (X-Gluc)] overnight at 37 °C. The leaves were destained using 70% ethanol and then photographed using a stereomicroscope (SZX16, Olympus, Tokyo, Japan).

### 4.12. Immunoprecipitation–Mass Spectrometry

Wheat cultivar YM49 plants were grown in a field at the Henan Agricultural University experimental station (Xuchang, China) in the 2020–2021 growing season. Grains from YM49 plants were harvested at 8, 16, 24, and 30 days after anthesis (daa), immediately frozen in liquid nitrogen, and stored at −80 °C until use. The grain samples were ground into powder using a mortar and pestle and then added to a ready-made IP buffer [100 mM Tris-HCl, 150 mM NaCl, 1 mM EDTA, 0.5% (*w*/*v*) NP-40, 1 × PMSF, 1 × protease inhibitor cocktail, 50 μM MG132 proteasome inhibitor, 2 mM DTT, and 5% Tween20]. The extracted protein solution and GS-specific antibody were incubated at 4 °C for 10 h. The beads were collected and washed three times with a wash buffer (50 mM Tris-HCl, 100 mM NaCl, 10% glycerol, and 0.05% TritonX-100). The SDS loading buffer was added, boiled, and rinsed three times. The supernatant was collected and subjected to SDS-PAGE denaturing gel electrophoresis. Trypsin was used to digest the gel strips containing bound proteins. The enzymatic digestion products were separated using high-performance liquid chromatography (HPLC) and then analyzed using a Q Exactive mass spectrometer (Thermo Fisher, USA). Identification and quantification analysis of proteins from the mass spectrometry data was performed using the MaxQuant 1.5.5.1 software.

### 4.13. Gene Ontology Annotation Analysis of TaGS Gene Family

Gene Ontology (GO) annotation was performed to investigate the potential functions of TaGS in biological processes. TaGS-interacting protein sequences were submitted to eggNOG (http://eggnog-mapper.embl.de/, accessed on 20 September 2025) for GO annotation, and TBtools were used for enrichment analysis and visualization.

## 5. Conclusions

In wheat, 12 TaGS genes were divided into four subfamilies: TaGS1-1, TaGS1-2, TaGS1-3, and TaGS2. These genes are distributed across subgenomes A, B, and D. The structure and protein sequences of the GS genes are highly conserved among wheat and five other representative species. All TaGS genes have experienced purifying selection, and both genome-wide and fragmental duplication events have contributed to the expansion of the TaGS gene family in wheat. The TaGS gene promoters may be regulated by various transcription factors and miRNAs. They also contain numerous cis-acting elements related to transcription, growth and development, plant hormones, and stress responses. The transcriptomic data and qRT-PCR assay revealed that the TaGS gene family exhibited differential expression levels in wheat under various abiotic stress conditions. For example, the expression levels of *TaGS1-3-4A/4B/4D* were significantly increased under cold, heat, salt, and drought stress conditions. This study lays a foundation for further functional research of TaGS genes in response to abiotic stress and may provide potential information for enhancing stress tolerance in wheat.

## Figures and Tables

**Figure 1 ijms-26-09403-f001:**
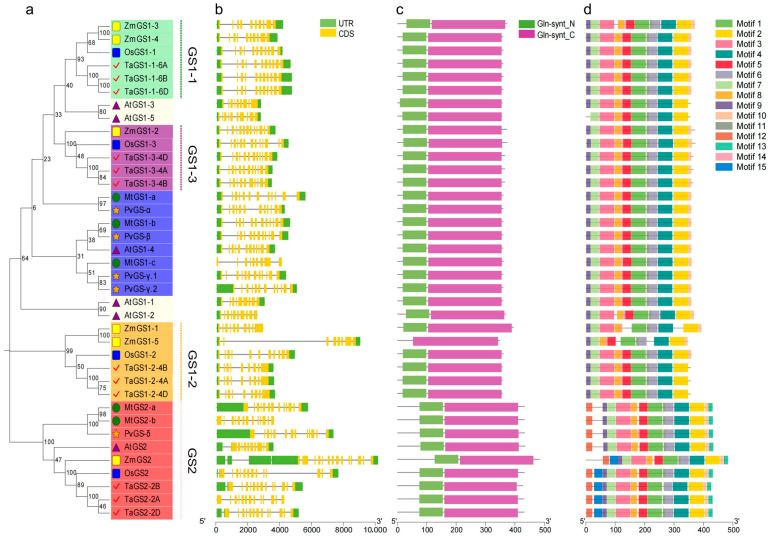
Evolutionary tree (**a**), gene structure (**b**), conserved domains (**c**), and conserved motif (**d**) analysis of TaGS genes. (**a**) The neighbor-joining (NJ) phylogenetic tree was constructed with protein sequences encoded by the longest transcript of each GS gene in *Taestivum* (Ta), *A. thaliana* (At), *O. Sativa* (Os), *Zea mays* (Zm), *Phaseolus vulgaris* (Pv), and *M. truncatula* (Mt) with bootstrap values of 1000 replicates. Different groups of GS proteins are distinguished by different colors. The different symbols indicated different species. (**b**) Exon/intron structures of TaGS genes from sweet orange. Green boxes, yellow boxes, and black lines represent untranslated regions (UTRs), exons, and introns, respectively. The lengths of the boxes and lines are scaled based on gene length. The exon and intron sizes are estimated using the scale at the bottom. (**c**) Conserved domain compositions of TaGS proteins in wheat. The green box represents Gln-synt_N, and the purple box represents Gln-synt_C. (**d**) Conserved motif distribution in TaGS protein. Different colored boxes represent different motif types.

**Figure 2 ijms-26-09403-f002:**
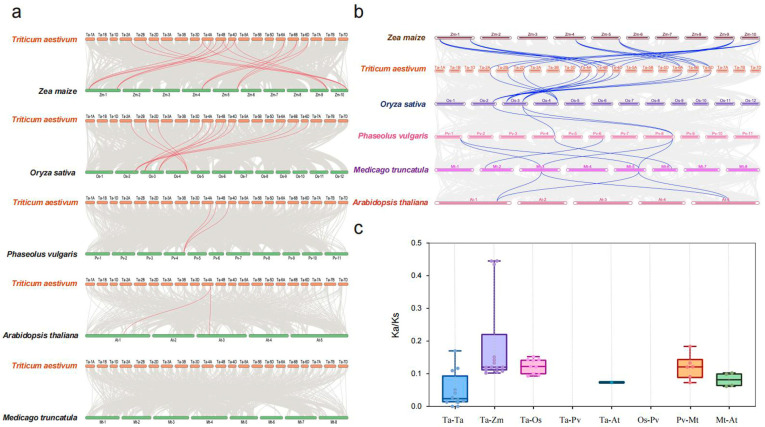
Collinearity and evolutionary analysis of GS gene. (**a**) The collinearity analysis between wheat and other five representative species. The gray lines in the background represent the collinear blocks within *Triticum aestivum* and other plant genomes, while the red lines highlight the syntenic TaGS gene pairs. (**b**) Homology analysis of the GS genes in wheat, maize, rice, common bean, Alfalfa, and Arabidopsis. (**c**) Comparison of the Ka/Ks values of GS homologous gene pairs among wheat, maize, rice, common bean, Alfalfa, and Arabidopsis.

**Figure 3 ijms-26-09403-f003:**
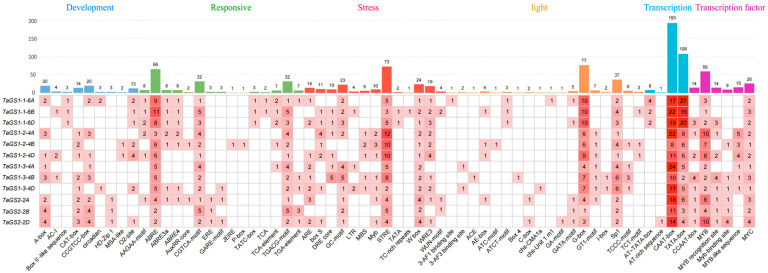
Classification of cis-acting elements in the members of the TaGS gene family in wheat. The different colors and numbers of grids indicated the numbers of different promoter elements. The histograms of different colors represented the sum of the cis-elements in each category.

**Figure 4 ijms-26-09403-f004:**
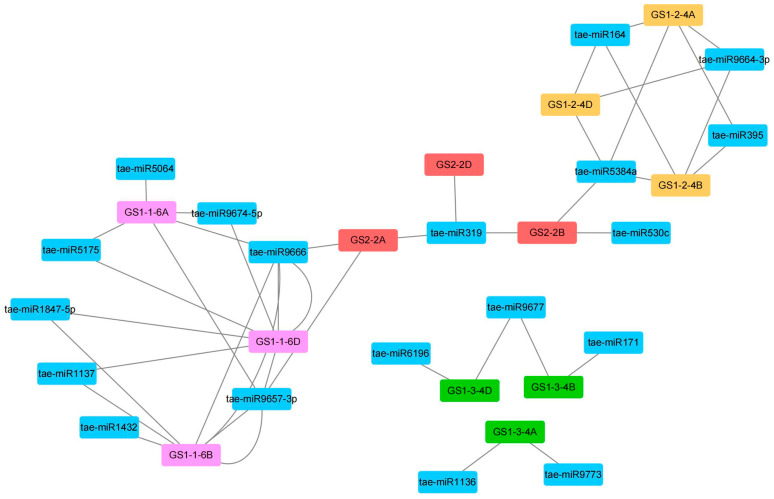
Regulatory network analysis of microRNAs and TaGS genes. Red represents GS2-2, rosy red represents GS1-1, yellow represents GS1-2, green represents GS1-3, and blue represents tae-miRNAs.

**Figure 5 ijms-26-09403-f005:**
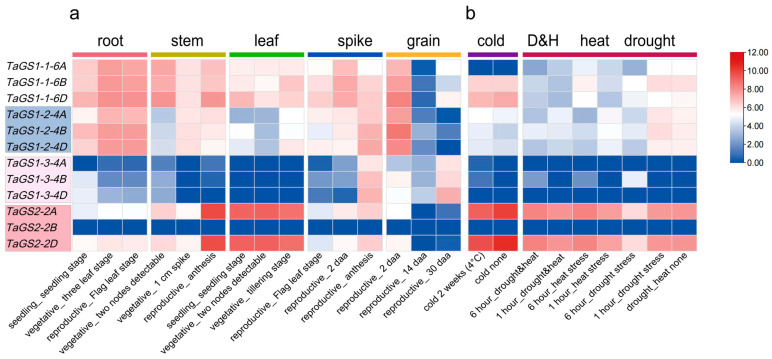
Heat map showing the expression profiles of the TaGS genes across various tissues and growth stages in wheat, derived from transcriptomic data. (**a**) Expression profiles of the TaGS genes in different wheat tissues. (**b**) Expression profiles of the TaGS genes in wheat under various abiotic stress conditions. The mean transcript levels (log_2_TPM) of each gene in all tissues analyzed were used to normalize the data.

**Figure 6 ijms-26-09403-f006:**
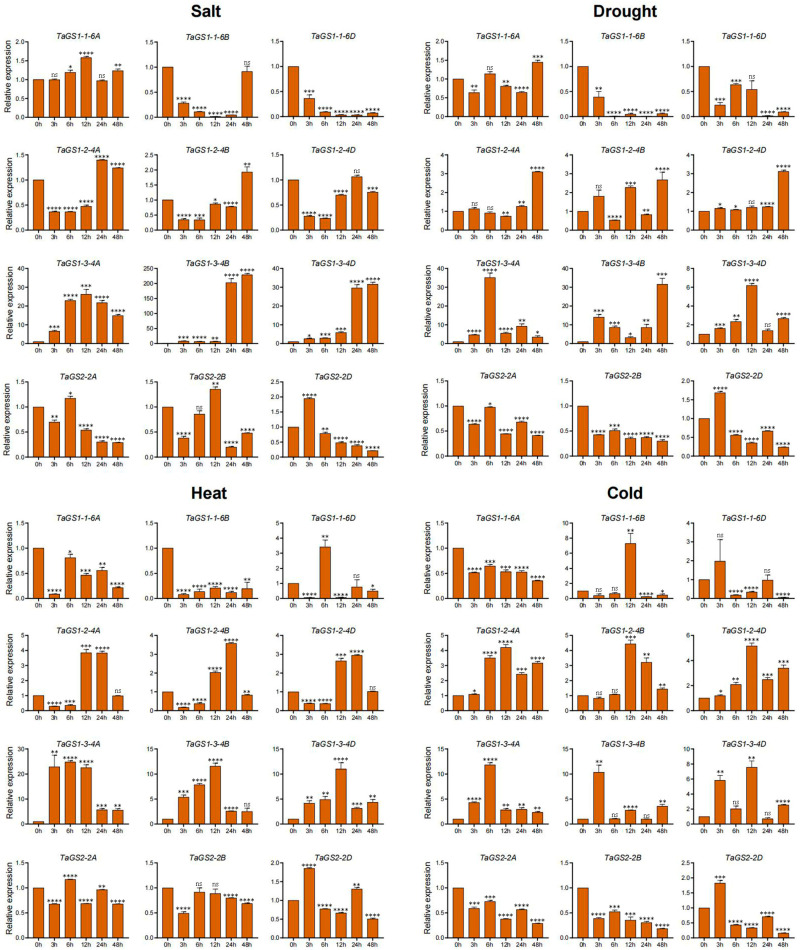
The relative expression levels of the TaGS genes in leaves under various abiotic stress conditions were detected by qRT-PCR. The wheat cultivar Yumai49 (YM49) seeds were surface-sterilized using 1% hydrogen peroxide solution for 10 min and rinsed five times with sterile distilled water. They were then placed on moistened filter paper in Petri dishes and germinated for two days at 28 °C in the dark. Fifteen wheat seedlings were transferred to a hydroponic device containing 1 L of Hoagland’s nutrient solution (pH 6.0) in a growth incubator at 22 °C and 65% humidity with long-day conditions. Hoagland’s nutrient solution was replaced every 3 days. After two weeks, consistent and uniform seedlings were transferred to heat (40 °C), cold (4 °C), salt (200 mM sodium chloride), and drought stress (20% m/V PEG 6000) conditions for treatment. The leaves of seedlings were harvested at 0 h, 3 h, 6 h, 12 h, 24 h, and 48 h after the stress treatment, respectively. The data were normalized to the internal control gene *TaACTIN*. The data are shown as means ± standard deviations (SDs) for three biological replicates. The data were analyzed using Microsoft Excel version 16. Statistical differences were determined using one-way ANOVA. * *p* < 0.05, ** *p* < 0.01, *** *p* < 0.001, and **** *p* < 0.0001.

**Figure 7 ijms-26-09403-f007:**
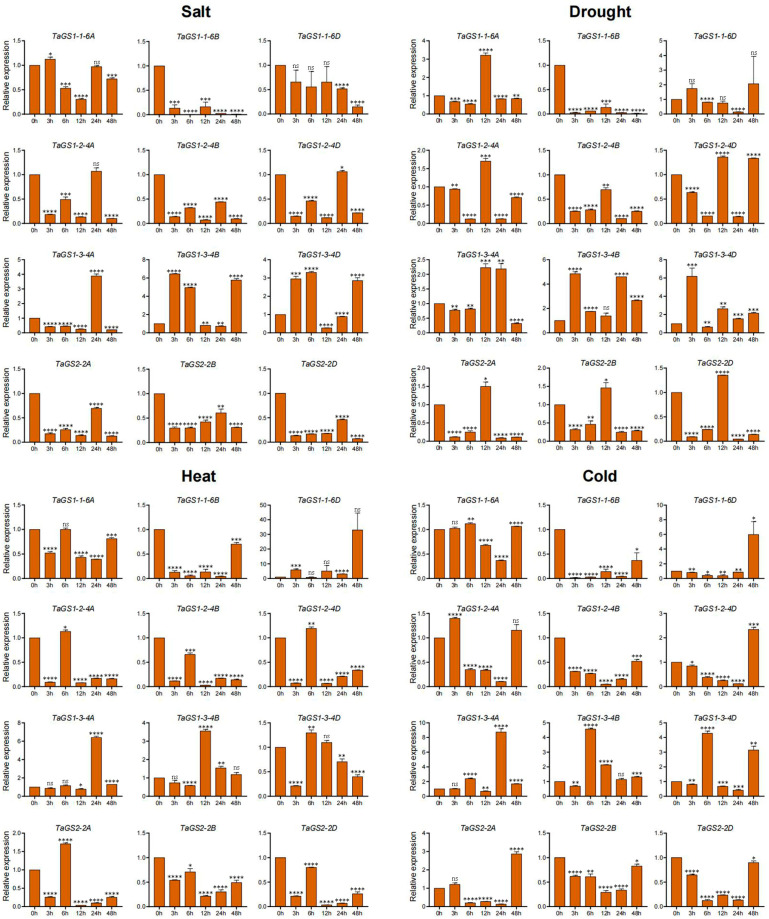
The relative expression levels of the TaGS genes in roots of wheat under four abiotic stress conditions using qRT-PCR. The wheat cultivar YM49 seeds were surface-sterilized using 1% hydrogen peroxide solution for 10 min and rinsed five times with sterile distilled water. They were then placed on moistened filter paper in Petri dishes and germinated for two days at 28 °C in the dark. Fifteen wheat seedlings were transferred to a hydroponic device containing 1 L of Hoagland’s nutrient solution (pH 6.0) in a growth incubator at 22 °C and 65% humidity with long-day conditions. Hoagland’s nutrient solution was replaced every 3 days. After two weeks, consistent and uniform seedlings were transferred to heat (40 °C), cold (4 °C), salt (200 mM sodium chloride), and drought stress (20% m/V PEG 6000) conditions for treatment. The roots of seedlings were harvested at 0 h, 3 h, 6 h, 12 h, 24 h, and 48 h after the stress treatment, respectively. The *TaACTIN* gene was used as the internal control. The data were normalized to the *TaACTIN* gene. The data are shown as means ± standard deviations (SDs) for three biological replicates. The data were analyzed using Microsoft Excel, version 16. Statistical differences were determined using one-way ANOVA. * *p* < 0.05, ** *p* < 0.01, *** *p* < 0.001, and **** *p* < 0.0001.

**Table 1 ijms-26-09403-t001:** The characterization of the TaGS gene family in wheat.

Gene Name	Gene ID	Number of AAs	MW (kDa)	pI	CL	Chr	Chromosome Location		Subcellular Localization
							Start	End	
*TaGS1-1-6A*	TraesCS6A02G298100.2	356	39.21	5.41	618,079,260	6A	531,394,366	531,398,363	Cyto
*TaGS1-1-6B*	TraesCS6B02G327500.1	356	39.21	5.41	720,988,478	6B	577,183,711	577,187,787	Cyto
*TaGS1-1-6D*	TraesCS6D02G383600LC.1	356	38.69	5.41	473,592,718	6D	462,483,422	462,487,522	Cyto
*TaGS1-2-4A*	TraesCS4A02G063800.1	354	38.73	5.45	744,588,157	4A	60,668,121	60,671,232	Cyto
*TaGS1-2-4B*	TraesCS4B02G240900.1	354	38.66	5.35	673,617,499	4B	499,898,695	499,901,767	Cyto
*TaGS1-2-4D*	TraesCS4D02G240700.1	354	39.61	5.34	509,857,067	4D	403,145,655	403,148,815	Cyto
*TaGS1-3-4A*	TraesCS4A02G266900.1	362	39.47	5.3	744,588,157	4A	579,428,158	579,431,177	Cyto
*TaGS1-3-4B*	TraesCS4B02G047400.1	362	39.48	5.66	673,617,499	4B	34,722,272	34,725,256	Cyto
*TaGS1-3-4D*	TraesCS4D02G047400.1	362	46.70	5.53	509,857,067	4D	22,946,578	22,949,866	Cyto
*TaGS2-2A*	TraesCS2A02G500400.1	427	46.08	5.75	780,798,557	2A	729,293,649	729,297,303	Chl
*TaGS2-2B*	TraesCS2B02G528300.1	423	46.70	5.89	801,256,715	2B	722,629,776	722,634,436	Chl
*TaGS2-2D*	TraesCS2D02G500600.1	427	46.08	5.75	651,852,609	2D	595,161,545	595,165,983	Chl

Note: AA, amino acid; MW, molecular weight; pI, isoelectric point; CL, chromosome length; Chr, chromosome; Cyto, cytoplasmic; Chl, chloroplast.

## Data Availability

All data supporting the conclusions of this study are included in this article and the Appendix A.

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
