# Peer review of "Characterization of the Glutamine Synthetase Gene Family in Wheat (*Triticum aestivum* L.) and Expression Analysis in Response to Various Abiotic Stresses"

_ijms, 2025, doi:10.3390/ijms26199403_

Round 1

Reviewer 1 Report

Comments and Suggestions for Authors

Dear Authors,

Reviewer comments ijms-3805086

The manuscript entitled „Characterization of the glutamine synthetase gene family in wheat (Triticum aestivum L.) and expression analysis in response to various abiotic stresses“ represents a useful study aimed at structural and functional characterization of the family of 12 TaGS genes in wheat cv. Yumai 49 by combining bioinformatics and expression analyses. I think that the manuscript provides complex information on TaGS gene family in wheat thus I can recommend it for publication in IJMS.

I have only a few minor comments on the present version of the manuscript which are provided below:

1/ Plant material: The source for wheat cv. Yumai 49 has to be given in Materials and methods section; i.e., it has to be specified from which institution (university, gene bank, breeding company, ) the seeds were obtained.

2/ Terminology: For PEG6000 treatment, I recommend to use the term „osmotic stress“ rather than „drought“ since drought means a water deficit in soil or other solid substrate while molecular mechanisms of the effects of Hoagland nutrient solution with added PEG6000 on the plants are different.

3/ Importance of PTMs in protein function modulation: I recommend the authors to cite some comprehensive reviews on the roles of PTMs in protein function modulations in plants exposed to stresses such as Vu et al. (2018) Protein language: Post-translational modifications talking to each other. Trends Plant Sci. 23(12), 1068-1080. https://doi.org/10.1016/j.tplants.2018.09.004 and Kosová et al. (2021) Plant proteoforms under environmental stress: Functional proteins arising from a single gene. Front. Plant Sci. 12, 793113. https://doi.org/10.3389/fpls.2021.793113

4/ Formal comments on the text related to English language and style:

Introduction, line 61: Replace the verb „was“ with „revealed“ in the statement: „Previous study showed that the expression level of TaGS genes revealed significant differences in roots of two wheat varieties…“

Final recommendation: Accept after a minor revision.

Author Response

1.Plant material: The source for wheat cv. Yumai 49 has to be given in Materials and methods section; i.e., it has to be specified from which institution (university, gene bank, breeding company, ) the seeds were obtained.

Response: The seeds of common wheat cultivar Yumai49 (YM49) in this study were produced by Pingan Seeds company.

2.Terminology: For PEG6000 treatment, I recommend to use the term“osmotic stress” rather than “drought” since drought means a water deficit in soil or other solid substrate while molecular mechanisms of the effects of Hoagland nutrient solution with added PEG6000 on the plants are different.

Response: In this study, we investigated the expression of TaGS in wheat seedling under drought stress simulated by 20% (m/v) polyethylene glycol (PEG6000) in a hydroponic culture according to a previous study (Tian et al., 2013). We sincerely thank the reviewer for this insightful and critical comment. In future research, we will conduct soil dehydration and rehydration experiments to investigate drought stress.

Reference:

Tian, F., Gong, J., Zhang, J., Zhang, M., Wang, G., Li, A., Wang, W. Enhanced Stability of Thylakoid Membrane Proteins and Antioxidant Competence Contribute to Drought Stress Resistance in the tasg1 Wheat Stay-Green Mutant. J Exp Bot. 2013, 64, 1509-20.

3.Importance of PTMs in protein function modulation: I recommend the authors to cite some comprehensive reviews on the roles of PTMs in protein function modulations in plants exposed to stresses such as Vu et al. (2018) Protein language: Post-translational modifications talking to each other. Trends Plant Sci. 23(12), 1068-1080. https://doi.org/10.1016/j.tplants.2018.09.004 and Kosová et al. (2021) Plant proteoforms under environmental stress: Functional proteins arising from a single gene. Front. Plant Sci.12, 793113. https://doi.org/10.3389/fpls.2021.793113

Response: We cited some references in revised manuscript.

 4.Formal comments on the text related to English language and style:

Introduction, line 61: Replace the verb “was” with “revealed” in the statement: “Previous study showed that the expression level of TaGS genes revealed significant differences in roots of two wheat varieties…”

Response: We replaced the verb was with revealed.

Reviewer 2 Report

Comments and Suggestions for Authors

The problem of increasing the protein content in wheat is an important and urgent task. The study of the GS gene family encoding glutamine synthetase seems to be an important and urgent task. The authors of the presented manuscript studied the GS gene family in wheat and compared it with other representatives of monocotyledonous and dicotyledonous plants. As a result, interesting comparative data were obtained. However, wheat is a complex object with different ploidy. What ploidy did the wheat variety used in this work have and was the dependence on ploidy of the composition of the GS family members and the number of conservative motifs noted?
116 - Figure in full and further in the text
238 - it is not clear from Table S8, what are 8,4,5 and 5 miRNAs?
282 - drought is indicated twice
293 - It should be noted that the authors not only actively used bioinformatics analysis, but also conducted an experiment on the effect of stress factors on the expression level of the TaGS genes. However, TaGS genes do not play a decisive role in stress resistance. They are sensitive to stress and, accordingly, affect protein biosynthesis and protein content
361 - a large experiment was conducted, but its idea is not entirely clear, since all proteins can interact with each other, they only differ in affinity.
437 - what kind of proteins are 14-3-3?
453 - what is Km? If this is the Michaelis constant, then free Gln cannot have it
Please provide statistical analysis in the Materials and Methods section.
Please provide references as required by the journal.

Author Response

The problem of increasing the protein content in wheat is an important and urgent task. The study of the GS gene family encoding glutamine synthetase seems to be an important and urgent task. The authors of the presented manuscript studied the GS gene family in wheat and compared it with other representatives of monocotyledonous and dicotyledonous plants. As a result, interesting comparative data were obtained. However, wheat is a complex object with different ploidy. What ploidy did the wheat variety used in this work have and was the dependence on ploidy of the composition of the GS family members and the number of conservative motifs noted?

Response: The common wheat cultivar Yumai49 (YM49, allhexaploid) used in this study is widely cultivated in the Yellow and Huai river valleys winter wheat zone of China. The seeds of YM49 were produced by Pingan Seeds company.

116 - Figure in full and further in the text

Response: We replaced Fig. with Figure in revised manuscript.

238 - it is not clear from Table S8, what are 8,4,5 and 5 miRNAs?

Response: We proved Table S8.
282 - drought is indicated twice

Response: We corrected this sentence.

293 - It should be noted that the authors not only actively used bioinformatics analysis, but also conducted an experiment on the effect of stress factors on the expression level of the TaGS genes. However, TaGS genes do not play a decisive role in stress resistance. They are sensitive to stress and, accordingly, affect protein biosynthesis and protein content.

Response: Our previous study show that overexpression of TaGS1/TaGS2 in tobacco enhance the tolerance of transgenic plants to drought stress (Yu et al, 2022), and Overexpression of TaGSr in Arabidopsis enhanced tolerance to low-nitrogen stress (Li et al., 2024). Other studies have demonstrated that overexpression OsGS2 in rice increased the tolerance of transgenic plants to salt and cold stress, and Overexpression of pine GS1 in poplar enhanced drought tolerance (Hoshida et al., 2000; El-Khatib et al., 2004). Based on these reports, we supposed that increasing GS expression and activity can enhance the plant tolerance to various abiotic stresses. There are three isoforms of TaGS1 (TaGS1;1, TaGS1;2 and TaGS1;3) encoded by nine genes, and TaGS2 is encoded by three genes in A, B and D chromosome respectively, but which GS1 or GS2 genes play key role in abiotic stress are still unknown in wheat. Therefore, the aim of this study is to identify the GS involved in various abiotic stresses using bioinformatics, transcriptomics and quantitative real-time PCR (qRT-PCR).

References:

El-Khatib, R.T., Hamerlynck, E.P., Gallardo, F., Kirby, E.G. Transgenic Poplar Characterized by Ectopic Expression of a Pine Cytosolic Glutamine Synthetase Gene Exhibits Enhanced Tolerance to Water Stress. Tree Physiol. 2004, 24, 729-36. 

Hoshida, H., Tanaka, Y., Hibino, T., Hayashi, Y., Tanaka, A., Takabe, T., Takabe, T. Enhanced Tolerance to Salt Stress in Transgenic Rice that Overexpresses Chloroplast Glutamine Synthetase. Plant Mol Biol. 2000, 43, 103-11. 

Yu, H., Zhang, Y., Zhang, Z., Zhang, J., Wei, Y., Jia, X., Wang, X., Ma, X. Towards Identification of Molecular Mechanism in Which the Overexpression of Wheat Cytosolic and Plastid Glutamine Synthetases in Tobacco Enhanced Drought Tolerance. Plant Physiol Biochem. 2020, 151, 608-20. 

Li, H., Yu, M., Zhu, X., Nai, F., Yang, R., Wang, L., Liu, Y., Wei, Y., Ma, X., Yu, H., Wang, X. TaGSr Contributes to Low-Nitrogen Tolerance by Optimizing Nitrogen Uptake and Assimilation in Arabidopsis. Environ Exp Bot. 2024, 219, 105657.

361 - a large experiment was conducted, but its idea is not entirely clear, since all proteins can interact with each other, they only differ in affinity.

Response: Previous studies suggested that TaGS1-2, TaGS1-3 and TaGS2-2Ab were associated with wheat grain protein content and thousand-grain weight. To verify whether TaGS forms complexes with other proteins to regulate wheat grain during grain-filling stages, 210 potential TaGS-interacting proteins in the grains were identified through an immunoprecipitation-mass spectrum (IP-MS) assay.

437 - what kind of proteins are 14-3-3?

Response: 14-3-3 proteins are phosphoserine-binding proteins that regulate the activities of a wide array of targets via direct protein-protein interactions. Plant 14-3-3 proteins bind a range of transcription factors and other signalling proteins, and have roles regulating plant development and stress responses.

453 - what is Km? If this is the Michaelis constant, then free Gln cannot have it

Response: Km is the Michaelis constant. We sincerely thank the reviewer for pointing out this critical inaccuracy in our terminology. We corrected the manuscript.

Please provide statistical analysis in the Materials and Methods section.

Response: We sincerely thank the reviewer for this constructive suggestion. We agree that a more detailed description of the statistical methods will enhance the clarity and reproducibility of our study. Accordingly, we have now added statistical analysis within the “Materials and Methods” section.

Please provide references as required by the journal.

Response: We have carefully proofread the entire reference list and in-text citations to ensure accuracy and completeness. We believe the revised manuscript now fully meets the IJMS's requirements.